# Environmental and Genetic Risk Factors in Developmental Dysplasia of the Hip for Early Detection of the Affected Population

**DOI:** 10.3390/diagnostics14090898

**Published:** 2024-04-25

**Authors:** Judit A. Ramírez-Rosete, Alonso Hurtado-Vazquez, Antonio Miranda-Duarte, Sergio Peralta-Cruz, Ramiro Cuevas-Olivo, José Antonio Martínez-Junco, Rosalba Sevilla-Montoya, Berenice Rivera-Paredez, Rafael Velázquez-Cruz, Margarita Valdes-Flores, Claudia Rangel-Escareno, Gerardo J. Alanis-Funes, Laura Abad-Azpetia, Sacnicte G. Grimaldo-Galeana, Monica G. Santamaría-Olmedo, Alberto Hidalgo-Bravo

**Affiliations:** 1Department of Genomics Medicine, National Institute of Rehabilitation (INRLGII), Calzada Mexico-Xochimilco 289, Arenal de Guadalupe, Mexico City 14389, Mexico; angelica.ram.rosete@gmail.com (J.A.R.-R.); ahvazquez03@gmail.com (A.H.-V.); antoniomirandaduarte@gmail.com (A.M.-D.); marvaldes@yahoo.com (M.V.-F.); anica.abd@gmail.com (L.A.-A.); a270012@alumnos.uaslp.mx (S.G.G.-G.); mgso82@hotmail.com (M.G.S.-O.); 2Department of Pediatric Orthopedics, National Institute of Rehabilitation (INRLGII), Calzada Mexico-Xochimilco 289, Arenal de Guadalupe, Mexico City 14389, Mexico; speralta@inr.gob.mx (S.P.-C.); ramirocuevas@yahoo.com (R.C.-O.); jmartinez@inr.gob.mx (J.A.M.-J.); 3Department of Genetics and Human Genomics, National Institute of Perinatology, Montes Urales 800, Lomas-Virreyes, Lomas de Chapultepec IV Secc, Miguel Hidalgo, Mexico City 11000, Mexico; rosalbasevilla@hotmail.com; 4Research Center in Policies, Population and Health, School of Medicine, National Autonomous University of Mexico, Zona Cultural s/n, CIPPS 2° Piso Ciudad Universitaria, Coyoacán, Mexico City 04510, Mexico; bereriveraparedez7@gmail.com; 5Genomics of Bone Metabolism Laboratory, National Institute of Genomic Medicine (INMEGEN), Arenal Tepepan, Tlalpan, Mexico City 14610, Mexico; rvelazquez@inmegen.gob.mx; 6Computational Genomics Department, Instituto Nacional de Medicina Genómica (INMEGEN), Arenal Tepepan, Tlalpan, Mexico City 14610, Mexico; crangel@inmegen.gob.mx; 7School of Engineering and Sciences, Tecnologico de Monterrey, Campus Querétaro, Querétaro 76130, Mexico; gerardo.alanisf@tec.mx

**Keywords:** developmental dysplasia of the hip, environmental risk factor, single nucleotide variant, congenital defect, early detection, genetic association

## Abstract

Diagnosis of developmental dysplasia of the hip (DDH) mostly relies on physical examination and ultrasound, and both methods are operator-dependent. Late detection can lead to complications in young adults. Current evidence supports the involvement of environmental and genetic factors, such as single nucleotide variants (SNVs). Incorporating genetic factors into diagnostic methods would be useful for implementing early detection and management of affected individuals. Our aim was to analyze environmental factors and SNVs in DDH patients. We included 287 DDH cases and 284 controls. Logistic regression demonstrated an association for sex (OR 9.85, 95% CI 5.55–17.46, *p* = 0.0001), family history (OR 2.4, 95% CI 1.2–4.5, *p* = 0.006), fetal presentation (OR 3.19, 95% CI 1.55–6.54, *p* = 0.002), and oligohydramnios (OR 2.74, 95%CI 1.12–6.70, *p* = 0.026). A model predicting the risk of DDH including these variables showed sensitivity, specificity, PPV, and NPV of 0.91, 0.53, 0.74, and 0.80 respectively. The SNV rs1800470 in *TGFB1* showed an association when adjusted for covariables, OR 0.49 (95% CI 0.27–0.90), *p* = 0.02. When rs1800470 was included in the equation, sensitivity, specificity, PPV and NPV were 0.90, 0.61, 0.84, and 0.73, respectively. Incorporating no-operator dependent variables and SNVs in detection methods could be useful for establishing uniform clinical guidelines and optimizing health resources.

## 1. Introduction

Developmental dysplasia of the hip (DDH) is one of the most common congenital defects. DDH comprises a spectrum of clinical manifestations, ranging from complete fixed dislocation of the joint to a discrete asymptomatic acetabular dysplasia in adults [1]. DDH preferentially affects females (female:male ratio of 5–8:1) and ~63% of the cases are unilateral, affecting the left hip in ~64% [2]. The incidence of DDH is highly variable among populations. Caucasians show an average incidence of 29.5 cases per 1000 live births, while in African populations, it can be as low as 3 per 1000 live births [2]. The reported incidence in Mexico ranges from 2–11 per 1000 live births, however, it is considered that there is an underestimation of this defect [3]. The variability of incidence may be partially related to the detection method and the age of the individuals.

Early diagnosis and treatment are fundamental for a favorable outcome in affected individuals. Follow-up studies have demonstrated that up to 46% of patients with late diagnosis and treatment will suffer osteoarthritis of the hip during young adulthood, leading to total hip replacement [4]. Diagnosis is generally suspected through physical examination (Barlow and Ortholani tests) and confirmed by ultrasound or radiographs [5]. Despite the available methods, there are many undiagnosed patients, who could suffer long-term complications. The high incidence of DDH and its long-term consequences have prompted the development of policies and strategies focused on early diagnosis in all countries. Screening programs have improved detection rates and, in the near future, it is expected that the number of surgeries related to DDH will decrease [6]. 

DDH can be isolated or associated with other birth defects such as congenital muscular torticollis, congenital talipes equinovarus, and spine and neuromuscular alterations. The presence of concomitant defects varies depending on the studied population (reviewed in [2]). 

The etiology of DDH involves environmental and genetic factors. Among the most relevant environmental risk factors are breech presentation, sex, family history, first-born and laterality [7]. On the other hand, the role of genetic factors is supported by high heritability values, ranging from 74% to 83% depending on the population studied [8]. In addition, the concordance between monozygotic and dizygotic twins is reported at 41% and 3%, respectively [9,10]. Furthermore, analyses of Caucasian families, where aggregation is observed, have revealed that first-degree relatives could have a risk as high as 12-fold for presenting DDH [11]. Different studies have reported genetic variants in families where DDH resembles a Mendelian inheritance pattern or where familial aggregation is present. Single nucleotide variants (SNVs) represent the focus of genetic research in DDH. Some reviews have gathered the associated variants across different populations [12,13]. More research in different populations is needed to unravel the role of these variants in DDH.

In Mexico, there are just a few reports regarding the environmental risk factors associated with DDH and none related to genetic factors. A better strategy for risk estimation can allow us to identify the newborns that need to be brought to a specific level of medical attention in order to diagnose DDH at an early stage and prevent long-term complications. The aim of this study was to evaluate the association of the clinical and environmental factors, along with previously described SNVs, in a series of patients with a confirmed diagnosis of DDH from Mexico. In addition, we performed next generation sequencing on genes that have been reported to have genetic variants in familial cases of DDH.

## 2. Materials and Methods

### 2.1. Study Population

A case–control study was conducted. Patients were recruited from the pediatric orthopedics department from March 2017 to February 2021. We assessed 571 children through physical examination looking for hip instability using the Ortolani and Barlow maneuvers. Children presenting positive hip instability underwent radiographic assessment. After radiographic assessment, DDH was confirmed in 287 children. Confirmed cases were classified according to the Tönnis classification [14]. A total of 284 children with no signs of hip instability were included as controls after DDH was excluded by hip radiography. 

Demographic data and risk factors were retrieved using a questionnaire. Environmental factors were included in the questionnaire based on previous publications [2,7]. The questionnaire included the following variables: sex of the participant, positive family history (this was established when a first- or second-degree relative of the participant was diagnosed with DDH), multiple pregnancy, weeks of gestation (WOG) (based on WOG, newborns were classified as term (38–42 WOG), preterm (<38 WOG) or postterm (>42 WOG)), fetal presentation (cephalic and other presentations), delivery mode divided as vaginal or cesarean section, presence of oligohydramnios, newborn’s weight and height, and mother’s height. This study was approved by the Institutional Ethics Committee (Approval 31/15) and all of the participants signed an informed consent form.

### 2.2. DNA Extraction and Single Nucleotide Variants Genotyping

Five ml of peripheral blood was collected from each participant in EDTA tubes. Total DNA was extracted using the Gentra Puregene Blood Kit (QIAGEN system Inc., Germantown MD, USA; Cat. No. 158467), according to the manufacturer’s instructions. Genotyping was performed through predesigned TaqMan Genotyping Assays (Thermo Fisher Scientific, Waltham MA, USA) following the manufacturer’s instructions. The following assays were used: for rs3732378 in *CX3CR1,* assay ID C_5687_1; for rs1800470 in *TGFB1*, assay ID C_22272997_10; for rs1569198 in *DKK1,* assay ID C_8742663_20, for rs224331 in *GDF5,* assay ID C__25619958_10 and for rs3744448 in *TBX4,* assay ID C__25804930_10. Real-time PCR was performed on a StepOne plus instrument, following the amplification protocol recommended for TaqMan probes (Applied Biosystems, Waltham MA, USA). Data were analyzed using StepOne software V2.3 (Applied Biosystems, Waltham MA, USA). Variant calling was obtained with the StepOne software V2.3 (Applied Biosystems, Waltham MA, USA).

### 2.3. Next Generation Sequencing

As a pilot study, looking for the presence of previously described genetic variants in familial cases of DDH, we carried out exome sequencing in 15 patients with confirmed DDH and three healthy controls using the SureSelect Human All Exon V7, which is designed to amplify 48.2 MB covering all known genes (Agilent, Santa Clara CA, USA, Cat. No. 5191-4004), following the manufacturer’s recommendations. Briefly, libraries were prepared using 500 ng of genomic DNA using the SSEL XT LI Lib Prep Kit (Agilent, Santa Clara CA, USA). DNA fragmentation was achieved using the Covaris S-220 instrument (Covaris, Woburn MA, USA). Samples were sequenced by paired-end sequencing in a NextSeq 550 System (Illumina, San Diego, CA, USA). WES Analysis Workflow follows a pipeline of best practices for variant calling in clinical sequencing. Raw sequence data in FASTQ format were aligned to the reference genome sequence using BWA-Mem. A binary alignment/map (BAM) file was then created within the SAMtools package [15]. The workflow used in this study is based explicitly on best practices for variant calling with the Broad Institute GATK [16]. This pipeline involves several steps to ensure that the alignment files are high-quality to guarantee variant calling accuracy. Quality control metrics for all fastq files were analyzed using FastQC and filtered with trimmomatic [17] before being aligned to the reference genome. The .sam file output from the alignment was converted to a compressed .bam file, marking the PCR duplicates. The sorting and indexing of the .bam file was performed with SAMtools and Picard. The GATK Haplotype-Caller conducted a base-quality score recalibration and local realignment around insertion–deletion sites and regions with poor mapping quality. In addition, variant calls were identified, and complex filtering also used the GATK HaplotypeCaller. Variants were annotated using ANNOVAR [18]. The RefGene database specifies known human protein-coding and non-protein-coding genes. The Clinvar_2021050 database was used to search for disease-specific variants. The dbnsfp41a database focuses on the functional prediction of variants in whole-exome data (this dataset already includes, among others, SIFT, PolyPhen2 HDIV, PolyPhen2 HVAR, MutationTaster, MutationAssessor scores), and finally, the gnomad exome and 1000 Genomes databases were used to determine the frequency of variants in the whole-exome data. Annotation files were converted to Mutation Annotation Format (MAF) files to analyze and visualize variants from large-scale sequencing studies [19]. Sequence analysis was focused on genes previously associated with familial cases of DDH: *HSPG2*, *ATP2B4*, *TGFB1*, *HOXD9*, *MMP24*, *UQCC1*, *GDF5*, *CX3CR1*, *IL6*, *ASPN*, *PAPPA2*, *HOXB9*, *TBX4*, *RETSAT*, *WISP3*, *BMP2K*, *DKK1*, and *PDRG1* [12,13].

### 2.4. Statistical Analysis

Data were analyzed using the Statistical Package for the Social Sciences version 19. Descriptive statistics are presented as means and standard deviations for quantitative variables and as percentages for qualitative variables. The Chi-square test was used for comparisons of qualitative variables. A *p* value < 0.05 was considered statistically significant. We developed an equation for predicting the risk of DDH, and the equation was evaluated through a multiple logistic regression model using the presence of DDH as the dependent variable. The odds ratios with their 95% confidence intervals (OR [95% CI]) were reported. Clinical variables considered in the model were selected based on previous reports, including, sex, family history, birth weight (>4000 gr), fetal presentation, oligohydramnios, and delivery mode [20]. The final prediction model included only the variables sex, family history, fetal presentation, and oligohydramnios, which showed statistical significance < 0.05. Afterward, we included the SNPs showing associations in the multivariate logistic regression model. This model met all of the assumptions (lack of strongly influential outliers, absence of multicollinearity, appropriate fitting, and specification of the model). The risk score from the final model to predict DDH was estimated as follows: risk score = 1/1 + e-riskscore. The accuracy of the risk score, derived from the model for predicting DDH, was evaluated through a ROC curve, and the optimal cut-point value was obtained using the Youden index [21]. In addition, we also estimated the sensitivity, specificity, positive predictive value (PPV), and negative predictive value (NPV). To test the performance of the equation, we estimated the capacity of the equation to distinguish between high and low-risk individuals considering the area under the curve (discrimination) [22,23].

Allele and genotype frequencies were obtained and Hardy–Weinberg equilibrium (HWE) was tested through Chi-square tests. For SNVs analysis, we considered four models on inheritance, dominant, recessive, additive and co-dominant as described elsewhere [24]. Analysis of the four models was conducted through a logistic regression model. In addition, allele frequencies were compared using Chi-square tests.

## 3. Results

### 3.1. Clinical and Environmental Factors Association Analysis

We included 287 children with a confirmed diagnosis of DDH and 284 children without evidence of DDH in the radiographs as controls. The age range of the total population study was 3–64 months. Among the patients with DDH, radiographs revealed that in 39.3%, the left and right hip were affected, while 38.1%, and 22.1% presented the defect only in the left or right hip, respectively. Considering the presence of subluxation or dislocation, we observed that in the right hip, 47.9% were an isolated dysplasia, whereas 14.9% and 37.2% were cases with subluxation or dislocation, respectively. Regarding the left hip, the percentage of cases with isolated dysplasia, subluxation and dislocation were 37.6%, 9.9%, and 52.5%, respectively. During the study period, 66.15% were treated with surgery, and 33.85% with a Pavlik harness.

In the group of cases, 84% were female and 16% male; the female:male ratio observed was 5:1. On the other hand, within the control group, 42% were females and 58% were males. Based on a literature review, we retrieved information regarding potential environmental risk factors associated with DDH development. Table 1 shows the comparison of these risk factors by study groups. Variables displaying statistically significant differences were sex, family history, oligohydramnios, fetal presentation, and mother’s height. We also performed a logistic regression multivariate analysis to identify the factors associated with the risk of presenting DDH (Table 1). The variables showing associations were female sex (OR 9.85, 95% CI 5.55–17.46, *p* = 0.0001), family history (OR 2.4, 95% CI 1.2–4.5, *p* = 0.006), fetal presentation (OR 3.19, 95% CI 1.55–6.54, *p* = 0.002), and oligohydramnios (OR 2.74, 95% CI 1.12–6.70, *p* = 0.026). The rest of the analyzed variables did not show a significant increase in the odds of DDH.

Interestingly, among the patients with a family history of DDH, we observed a pair of monozygotic twins where both children were affected. In addition, in two families, the genealogy resembled Mendelian inheritance patterns. In one family, there were affected males and females and vertical transmission, compatible with autosomal dominant inheritance (Appendix A). In another family, we observed only affected women. In this family, there was vertical transmission from an affected woman to an affected woman and from an unaffected male to an affected woman. However, there was no defined Mendelian inheritance pattern (Appendix A).

### 3.2. Single Nucleotide Variant Association Analysis

Based on a literature review looking at articles demonstrating associations of SNVs with DDH in different populations, we selected five SNVs for genotyping in the study population. Table 2 depicts the SNVs details, including minor allele frequencies (MAFs) according to the 1000 genomes database, allele frequencies of the current study population, and HWE assessment through Chi-square tests.

Table 3 reports the allele and genotype frequencies distribution across the study population. There were no significant differences in the distribution of allele or genotype frequencies between the study groups. Each variant was analyzed using a logistic regression model considering four inheritance patterns, dominant, recessive, co-dominant, and additive.

Initially, we performed an univariate analysis (Table 4), and with this analysis only the rs224331 in *GDF5* showed a significant association under the recessive and co-dominant model (OR 1.50, 95% CI 1.01–2.24, *p* = 0.04; and OR 0.63, 95%CI 0.42–0.94, *p* = 0.02, respectively) (Table 4). Afterward, we adjusted the logistic regression model by the environmental variables showing an association, namely, sex, family history, fetal presentation, and oligohydramnios (Table 5). After adjustment, the association of the variant rs224331 in *GDF5* was no longer observed. Nevertheless, the rs1800470 in *TGFB1* showed a significant association under the recessive model after adjustment for covariables (OR 0.49; 95% CI 0.27–0.90, *p* = 0.02).

### 3.3. Generation of Prediction Models

Genetic variants are expected to contribute to the identification of populations at a higher risk of suffering multifactorial conditions such as DDH. Based on our results, we tested the capacity of the environmental variables (sex, family history, fetal presentation, and oligohydramnios) in conjunction with the variant rs1800470 in TGFB1, under the recessive model, to predict the occurrence of DDH. We proceeded with the generation of receiving operating characteristic (ROC) curves to determine the sensitivity and specificity of the selected environmental and genetic variables. Considering only the environmental variables in the model, the area under the curve (AUC) was 0.79 (Figure 1A). The sensitivity and specificity were 0.91 and 0.53, respectively, with a PPV and NPV of 0.74 and 0.80, respectively. When the rs1800470, considering the recessive model, was added to the model, the AUC increased to 0.811 (Figure 1B). Furthermore, the sensitivity and specificity were 0.90 and 0.61 respectively, while the PPV and NPV were 0.84 and 0.73. The other SNPs were not tested in this analysis because they did not reach statistical significance under the adjusted linear regression model.

### 3.4. Next Generation Sequence Analysis

We carried out NGS focused on 18 genes previously described as carrying variants possibly associated with familial cases of DDH. Three control individuals and 15 patients were sequenced. Details of the variants found in the analyzed individuals are provided in Appendix A. We found a total of 287 and 202 different variants in the groups of patients and controls, respectively (Appendix A). Variants were found in the 18 analyzed genes. All variants were found in a heterozygous state, and allele frequencies within the group of cases were calculated based on the 30 alleles present in the 15 cases sequenced. In addition, we obtained the MAF from gnomAD and the CADD score for each variant. The CADD score is a known tool for assessing a variant’s deleteriousness, and rather than providing a categoric classification it can effectively prioritize variants [25]. From the 287 different variants, 148 were exclusively found in the group of cases and the remaining 139 existed in the groups of controls and cases. From the variants found only in the group of cases, none of them have been previously published in association with DDH. Furthermore, 6 out of these 148 variants have a reported MAF < 0.01 in gnomAD and a CADD score above 20, and all of them are in the HSPG2 gene. From the 139 variants shared between cases and controls, ten have been previously published in association with DDH. These were found in the genes *CX3CR1*, *DKK1*, *GDF5*, *HOXB9*, *HOXD9*, *PAPPA2*, *TGFB1* and *WISP3*.

## 4. Discussion

Developmental dysplasia of the hip (DDH) is one of the most prevalent congenital defects affecting populations worldwide. DDH is a multifactorial disease and this implies an interplay between environmental and genetic factors, which can vary between different populations. This study analyzed previously acknowledged environmental and genetic risk factors associated with DDH. Our results showed that sex, family history, fetal presentation and oligohydramnios were the most relevant environmental risk factors in the study population. Regarding the genetic factors, we found an association of DDH with the SNV rs1800470 in *TGFB1.*

DDH is one of the conditions for which early detection can lead to an enormous benefit for the patient and health systems. There are a few studies analyzing the economic cost of screening programs and treatment of affected individuals. In the United Kingdom, a study conducted over an 11-year period analyzed the cost-effectiveness of ultrasound screening (USS) and subsequent treatment, demonstrating the effectiveness of the USS in early detection and a reduction of treatment costs [26]. Unfortunately, data regarding the annual budget dedicated to treating DDH patients in Mexico is limited. A study published in 2011 estimated a total cost of $16,600 USD per treated patient. At that moment, the demographic data calculated there were nearly 2 million children aged three months and the expected number of cases was 12,000 [3]. Furthermore, based on the observations in other countries, early detection is crucial for optimizing human and economic resources [6]. Therefore, it is necessary to implement health policies to promote early detection.

Despite being a common congenital defect, detection and treatment guidelines differ between countries [2]. Environmental factors have proven to be useful for the suspicion of the population at risk. These factors can be retrieved during a medical interview and can alert the clinician about the possibility of DDH [13]. The Barlow and Ortholani tests are a widely used initial screening approach; nevertheless, different studies have shown that these maneuvers can have a false-negative rate ranging from 7 up to 19.5% [27,28]. In addition, the sensitivity of a physical examination ranges from 60 to 67%. It is important to bear in mind that physical examination relies on the training and experience of the physician. Therefore, the sole practice of the Barlow and Ortholani maneuvers should be undertaken with caution for a diagnosis of DDH. On the other hand, ultrasound has shown a high specificity and sensitivity, with an acceptable inter-operator variability [29]. However, in some countries like Mexico, the cost and availability of ultrasound can be a limiting factor, especially in rural communities [30]. So far there is no universal consensus regarding the method or methods for a diagnosis of DDH. Recently, efforts have been directed to unravel the genetic factors involved in the etiology of DDH. Several genome-wide association studies as well as targeted pathogenic variants analysis have been conducted recently. These studies have found an association of several variants in genes related to osteogenesis and joint formation, among other relevant biological processes [13,31].

Herein, we present a cohort of patients with a confirmed diagnosis of DDH through radiographic studies. The environmental and demographic factors investigated were selected based on previous reports [2,7]. In 2020, Roposch and colleagues described a prediction model including four relevant factors, sex, first-degree family history of DDH, birthweight > 4000 g, and abnormal examination of the hip. According to their model, females (OR = 5.6, 95% CI, 2.9–10.9, *p* < 0.001) with a positive family history of DDH (OR = 4.5, 95% CI, 2.3–9.0, *p* < 0.0001) birthweight above 4000 gr (OR = 1.6, 95% CI, 0.6–4.2, *p* = 0.34), and with a positive DDH clinical examination (OR = 58.8, 95% CI, 31.9–108.5 *p* < 0.001) are the ones with the highest risk of DDH. Their model discriminated well between newborns with and without DDH (C statistic = 0.9, 95% CI, 0.8–0.9, goodness-of-fit *p* = 0.35) [20]. A previous report from Guanajuato, Mexico analyzed 100 patients from public and private hospitals. In this study, the risk factors with a significant OR were obstetric presentation (OR 5.32, 95% CI 1.76–16.13) and excessive swaddling (OR 4.91, 95% CI 1.90–12.66) [32]. Interestingly, family history did not show a significant risk in this group. It is worth mentioning that they did not include sex in the analysis, which has been identified as one of the main risk factors across several populations. In our study population, firstly, we identified the environmental factors with significant differences between cases and controls. We found that the most relevant variables for predicting the risk of DDH were sex, positive family history, fetal presentation, and oligohydramnios.

Previous studies have identified SNVs associated with DDH in different populations. Among the most relevant we selected five for testing in our study population, rs3732378 in CX3CR1, rs18004 in TGFB1, rs1569198 in DKK1, rs224331 in GDF5 and rs3744448 in TBX4. Allele and genotype distribution did not differ between cases and controls. We analyze the association of these SNVs through a logistic regression model, initially unadjusted and then adjusted considering the variables sex, family history, fetal presentation, and oligohydramnios. The unadjusted analysis showed an association only with the rs224331 in GDF5 under the recessive and co-dominant inheritance models. Nevertheless, the association was no longer observed after adjusting for covariables. Different SNVs in GDF5 have been associated with DDH in Chinese, European, and Saudi Arabian populations [33,34,35]. GDF5 has a recognized role in bone and joint development [13]. Recently, Chen, et al., have shown that the GDF5 locus contains many separate regulatory elements that control expression of the gene at different joint sites [36]. Furthermore, recent evidence suggests that hypermethylation of the GDF5 promoter, leading to decreased expression, could be involved in DDH etiology [37]. 

After adjustment for covariables, the SNV rs1800470 in TGFB1 was the only one showing a significant association. rs1800470 has been found in association with osteoarthritis (OA) of the hip secondary to DDH in the adult population of Croatia and Turkey [38,39,40]. Furthermore, a study in a cohort of the Chinese population identified a significant association of this variant with DDH. Interestingly, when patients with DDH were stratified by severity, the association remained only in the group with the most severe clinical presentation of DDH [41].

To determine the ability for discriminating the affected individuals, we generated ROC curves. First, we consider a model using the four environmental variables showing a significant association (sex, family history, fetal presentation, and oligohydramnios). This model showed an AUC of 0.79 and sensitivity and specificity were 0.91 and 0.53, respectively, with a positive predictive value (PPV) and negative predictive value (NPV) of 0.74 and 0.80, respectively. Afterward, we included the SNV rs1800470 under the recessive model in the analysis. This model showed the AUC increased to 0.811; furthermore, the sensitivity and specificity were 0.90 and 0.61, respectively, while the PPV and NPV were 0.84 and 0.73. While the inclusion of the rs1800470 under the recessive model showed an increase of AUC to 0.811, it is noteworthy that the improvement may be considered modest. The clinical interpretation of this difference requires careful evaluation, as values close to 0.80 suggest moderate discrimination. The inclusion of genetic factors, such as rs1800470, may provide valuable information, but as observed, its contribution to the model did not result in a substantial improvement in predictive capacity. Future studies could explore the inclusion of more genetic variants and assess their impact on the accuracy of the predictive model for the development of developmental dysplasia of the hip.

Several methods have been proposed for the screening and early diagnosis of DDH, and among the most used worldwide are the Barlow and Ortholani maneuvers and ultrasound. However, these methods have an operator-dependent component leading to variation of the detection rates as mentioned above. The incorporation of non-operator dependent elements to the diagnostic methods can reduce bias and improve detection rates. Here, we explored factors that are not operator-dependent. The four environmental factors included in the model can be easily obtained during a medical interview or pregnancy follow-up. The risk of a misinterpretation of any of these four variables is minimal. Additionally, the incorporation of SNVs added another factor that has also a minimum risk of misinterpretation. Having an unbiased screening method can help us identify the high-risk individuals that need immediate action to diagnose DDH and prioritize them for early treatment. With these actions, the economic burden on health systems can change from attending to the repercussions of a delayed diagnosis and treatment to identifying patients in need of specialized attention.

Among the study population, we observed one family resembling an autosomal dominant inheritance pattern. There are some reports presenting families with affected individuals in more than one generation. Most of those publications have been useful for identifying genetic variants associated with DDH. However, some of the identified variants have also been found in healthy controls, ruling out a causative role and suggesting just an association [42,43,44]. Molecular analysis of those families could provide valuable information regarding genetic variants associated with DDH in Mexican population. Nevertheless, that aim is beyond the scope of the present work and will be the objective of future analysis. Some studies have found SNVs of interest after analyzing familial cases of DDH; segregation of these variants in affected individuals suggests a major role in the pathogenesis of the condition [31]. Nevertheless, more evidence is needed to consider the discovered variants as factors in the causal chain of events leading to DDH. The NGS analysis performed in our 15 patients with DDH revealed variants exclusively in the group of cases in the 18 analyzed genes. From the different 287 variants identified, ten have been previously associated with DDH, but these ten variants were found in cases and controls in our study population [12,13,31]. Among the 148 variants found exclusively in the group of cases, six have a MAF < 0.01 and a CADD > 20. Allele count in population databases such as gnomAD is a valuable resource for unravelling variants involved in the expression of a determined phenotype [45]. The CADD score was also considered because it is able to reveal a potential deleterious effect of the variant on gene function. There is no cutoff value for the CADD score, but the highest scores point toward a negative effect of the variant on gene function [25]. The list of variants presented here needs to be studied in depth to determine if they can have clinical value in the assessment and management of patients with DDH.

Our study has some limitations. Only some of the patients had X-ray images obtained at our institute because we do not assess newborns; therefore, we rely on the interpretation of the referral center. Nevertheless, orthopedists are trained in detecting DDH since it is one of the most common congenital defects. Furthermore, this was a single-center study, with patients living mainly in the metropolitan area of Mexico City. Consequently, caution should be exercised when generalizing our results to broader populations. Furthermore, it is crucial to acknowledge the potential for population stratification as an additional limitation. Our study primarily encompassed patients residing in the metropolitan area of Mexico City, and the ethnic and socio-demographic composition may not fully represent the diversity found in broader populations. Population stratification can introduce confounding variables that may impact the generalizability of our findings to other ethnic or geographic groups. While efforts were made to control for known factors, such as family history and environmental variables, the potential influence of unmeasured factors related to population structure remains a consideration. We have a relatively small sample size and some missing data during the follow-up. We are aware that a larger control group could strengthen the results. Nevertheless, our study has some strengths. First, we did not include patients with other congenital defects such as skeletal dysplasia, talipes equinovarus or neuromuscular diseases, avoiding a spurious association. Second, the National Institute of Rehabilitation is a national center for DDH reference, assuring the expertise and training of the medical staff.

## 5. Conclusions

In conclusion, our study has identified environmental factors such as sex, family history, fetal presentation, and oligohydramnios that are significantly associated with the risk of DDH in the Mexican population. Additionally, we explored the role of specific genetic variants, such as rs1800470 in TGFB1, although their contribution to the predictive model was modest. Our findings provide a valuable basis for the early identification of individuals at risk of DDH. Further studies with larger samples and the inclusion of more genetic variants are suggested to refine the predictive models. Furthermore, advocating for the implementation of health policies that promote early detection of DDH, with particular attention to the factors identified in this study, is crucial for optimizing medical resources and enhancing long-term outcomes for patients.

## Figures and Tables

**Figure 1 diagnostics-14-00898-f001:**
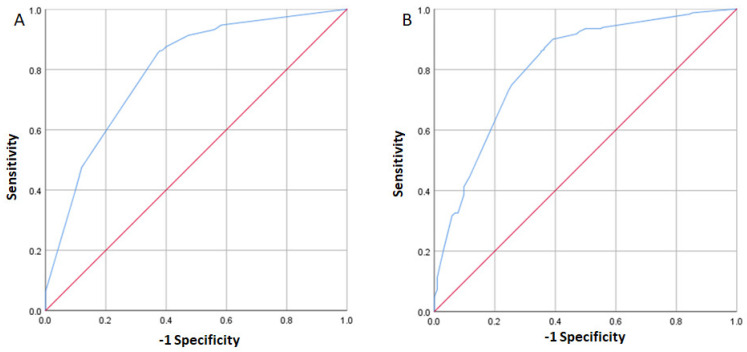
ROC curves. Analysis including the variables sex, family history, fetal presentation, and oligohydramnios, AUC 0.79 (**A**), subsequent inclusion of rs1800470 in TGFB1 in the model, AUC 0.811 (**B**).

**Table 1 diagnostics-14-00898-t001:** Demographic and clinical characteristics of the study population.

Variable	Categories	Control 284 N (%)	Case 287 N (%)	*p*-Value	OR (95% CI)	*p*-Value
Sex	Male	166 (58)	45 (16)	0.0001 *	Ref.	
	Female	118 (42)	242 (84)		9.85 (5.55–17.46)	0.0001
Family History	No	155 (55)	204 (71)	0.002 *	Ref.	
	Yes	27 (10)	77 (27)		2.4 (1.2–4.5)	0.006
Multiple pregnancy	No	174 (61)	275 (96)	0.097 *	Ref.	
	Yes	8 (3)	5 (2)		0.577 (0.06–5.02)	0.619
WOG	38–42	146 (51)	227 (79)	0.896 *	Ref.	
	<38	27 (28)	43 (15)		0.94 (0.40–2.16)	0.884
	>42	8 (3)	10 (3)		0.84 (0.21–3.38)	0.814
Delivery	Vaginal	86 (30)	107 (37)	0.052 *	Ref.	
	Cesarean	95 (33)	172 (60)		1.25 (0.71–2.21)	0.425
Oligohydramnios	No	165 (58)	223 (78)	0.004 *	Ref.	
	Yes	16 (6)	51 (18)		2.74 (1.12–6.70)	0.026
Fetal presentation	Cephalic	152 (54)	195 (68)	0.001 *	Ref.	
	Other	24 (8)	72 (25)		3.19 (1.55–6.54)	0.002
Newborn Weight (Kg) ^#^		3.01 (0.60)	3.04 (0.56)	0.60 **	1.21 (0.68–2.14)	0.513
Newborn Height (cm) ^#^		49.17 (4.3)	49.77 (3.22)	0.12 **	0.74 (0.03–14.75)	0.84
Mother’s Height (cm) ^#^		157.96 (6.1)	156.09 (6.2)	0.03 **	0.04 (0.001–4.16)	0.181

^#^ Quantitative variables presented as mean (standard deviation). * *p* values were obtained through the Chi-squared test. ** *p* values were obtained through Student’s *t*-test. WOG, weeks of gestation. Ref., category considered as the reference for the odds ratio (OR) calculation. 95% CI: 95% Confidence interval.

**Table 2 diagnostics-14-00898-t002:** Description of the analyzed SNVs and reported MAF according to the 1000 Genomes Project Phase 3.

Gene	SNV	RA	AA	Location GRCh38	MAF GBL ^1^	MAF CEU ^2^	MAF MXL ^3^	MAF INR ^4^	HWE *p*-Value
*TGFB1*	rs1800470	G	A	chr19:41353016	0.45	0.38	0.45	0.47	0.79
*CX3CR1*	rs3732378	G	A	chr3:39265671	0.09	0.17	0.23	0.21	0.19
*TBX4*	rs3744448	G	C	chr17:61456507	0.18	0.16	0.28	0.22	0.55
*GDF5*	rs224331	C	A	chr20:35434589	0.38	0.36	0.2	0.13	0.21
*DKK1*	rs1569198	A	G	chr10:52316511	0.32	0.49	0.34	0.28	0.15

MAF, minor allele frequency. RA, reference allele. AA, alternative allele. ^1^ Global population. ^2^ Utah Residents with ancestry from North and West Europe. ^3^ Mexican population living in Los Angeles, CA, USA. ^4^ Current population study.

**Table 3 diagnostics-14-00898-t003:** Allele and genotype distribution among study groups.

Gene	SNP	Genotype/ Allele	Control N (%)	Case N (%)	*p*-Value
*TGFB1*	rs1800470	GG	33 (19)	61 (25)	0.145
		GA	82 (48)	125 (50)	
		AA	57 (33)	62 (25)	
		G	148 (43%)	247 (50%)	0.053
		A	196 (57%)	249 (50%)	
*CX3CR1*	rs3732378	GG	171 (63)	174 (63)	0.986
		GA	86 (32)	88 (32)	
		AA	15 (6)	15 (6)	
		G	428 (79%)	436 (79%)	0.99
		A	116 (21%)	118 (21%)	
*TBX4*	rs3744448	GG	152 (58)	177 (65%)	0.247
		GC	94 (36)	81 (30%)	
		CC	14 (5)	13 (5%)	
		G	398 (77%)	435 (80%)	0.11
		C	122 (23%)	107 (20%)	
*GDF5*	rs224331	CC	2 (1%)	4 (1%)	0.066
		CA	72 (29%)	55 (20%)	
		AA	177 (71%)	213 (78%)	
		C	76 (15%)	63 (12%)	0.09
		A	426 (85%)	481 (88%)	
*DKK1*	rs1569198	AA	95 (51%)	109 (51%)	0.636
		AG	79 (42%)	93 (44%)	
		GG	14 (7%)	11 (5%)	
		A	269 (72%)	311 (73%)	0.64
		G	107 (28%)	115 (27%)	

*p*-value was obtained through Chi-square test.

**Table 4 diagnostics-14-00898-t004:** Association analysis considering unadjusted inheritance models.

Gene/Variant	Dominant		Recessive		Co-Dominant		Additive	
	OR (95% CI)	*p*-Value	OR (95% CI)	*p*-Value	OR (95% CI)	*p*-Value	OR (95% CI)	*p*-Value
*TGFB1/*rs1800470	0.72 (0.45–1.17)	0.19	0.67 (0.43–1.03)	0.06	1.11 (0.75–1.64)	0.58	0.82 (0.49–1.36) 0.58 (0.33–1.02)	0.54 0.06
*CX3CR1*/rs3732378	0.97 (0.68–1.37)	0.87	0.96 (0.46–2.00)	0.91	0.98 (0.68–1.40)	0.91	0.97 (0.67–1.40) 0.95 (0.45–2.01)	0.89 0.90
*TBX4/*rs3744448	0.74 (.52–1.05)	0.09	0.88 (.40–1.91)	0.75	0.74 (.52–1.07)	0.11	0.73 (0.50–1.06) 0.79 (0.36–1.738)	0.10 0.56
*GDF5*/rs224331	0.53 (0.09–2.96)	0.47	1.50 (1.01–2.24)	0.04	0.63 (0.42–0.94)	0.02	0.38 (0.06–2.16) 0.60 (0.10–3.32)	0.27 0.56
*DKK1*/rs1569198	0.97 (0.65–1.44)	0.89	0.67 (0.29–1.52)	0.34	1.06 (0.71–1.59)	0.74	1.02 (0.68–1.54) 0.68 (0.29–1.58)	0.90 0.37

**Table 5 diagnostics-14-00898-t005:** Association analysis considering adjusted inheritance models.

Gene/Variant	Dominant		Recessive		Co-Dominant		Additive	
	OR (95% CI)	*p*-Value	OR (95% CI)	*p*-Value	OR (95% CI)	*p*-Value	OR (95% CI)	*p*-Value
*TGFB1*/rs1800470	0.692 (0.33–0.419)	0.31	0.49 (0.27–0.90)	0.02	1.45 (0.82–2.549)	0.19	0.89 (0.41–1.93) 0.45 (0.20–1.03)	0.78 0.06
*CX3CR1*/rs3732378	0.81 (0.50–1.31)	0.39	0.73 (0.28–1.87)	0.51	0.87 (0.52–1.43)	0.58	0.83 (0.50–1.39) 0.68 (0.26–1.799	0.49 0.44
*TBX4*/rs3744448	0.75 (0.46–1.22)	0.24	0.90 (0.32–2.529)	0.84	0.75 (0.45–1.24)	0.27	0.74 (0.44–1.23) 0.81 (0.28–2.30)	0.25 0.69
*GDF5*/rs224331	0.49 (0.07–3.25)	0.46	1.422 (0.81–2.47)	0.21	0.62 (0.35–1.08)	0.09	0.33 (0.04–2.28) 0.55 (0.08–3.71)	0.26 0.54
*DKK1*/rs1569198	0.76 (0.43–1.32)	0.33	0.54 (0.17–1.70)	0.29	0.87 (0.49–1.52)	0.62	0.81 (0.45–1.44) 0.49 (0.15–1.61)	0.47 0.24

Logistic regression model was adjusted by the co-variables sex, family history, fetal presentation, and oligohydramnios.

## Data Availability

The original data presented in the study are openly available in (pending).

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
