# Peer review of "Environmental and Genetic Risk Factors in Developmental Dysplasia of the Hip for Early Detection of the Affected Population"

_diagnostics, 2024, doi:10.3390/diagnostics14090898_

Round 1

Reviewer 1 Report

Comments and Suggestions for Authors

 The manuscript ”Environmental and genetic risk factors in developmental dysplasia of the hip for early detection of the affected population” submitted by Ramírez-Rosete etc identified risk factors such as sex, family history, fetal presentation, oligohydramnios and specific genetic variants that are significantly associated with the risk of DDH in the Mexican population. Although this manuscript is interesting, there are still some questions that need to be addressed and further verification.

1. In clinical work, DDH risk factors which is considered in diagnosis of DDH at an early stage is needed to be addressed.

2. This study lacks certain novelty, among these risk factors including environmental factors and genetic varies which one is their new findings.

3. In the manuscript, the authors suggest sex and family history as environmental factors, but no detailed descreptions were provided, the author need to explain.

4. To further validate the findings, more samples from multiple centers are needed.

5. In addition, the study did not provide a detailed description of experiment in 2.2 and 2.3 for DNA extraction and single nucleotide variants genotyping, Next Generation Sequencing.

Reviewer 2 Report

Comments and Suggestions for Authors

Line 23: diagnosis of what? the abstract intro si generic

Please specify the diagnosis you performed for DDH

I cannot find age in the results

in discussion, please include a short resume at the very beginning

Round 2

Reviewer 2 Report

Comments and Suggestions for Authors

Concerns addressed, it can be published